# Nuclear spin assisted quantum tunnelling of magnetic monopoles in spin ice

C. Paulsen[1], S.R. Giblin[2], E. Lhotel[1], D. Prabhakaran[3], K. Matsuhira [4], G. Balakrishnan [5] & S.T. Bramwell[6]

Extensive work on single molecule magnets has identified a fundamental mode of relaxation arising from the nuclear-spin assisted quantum tunnelling of nearly independent and quasi-classical magnetic dipoles. Here we show that nuclear-spin assisted quantum tunnelling can also control the dynamics of purely emergent excitations: magnetic monopoles in spin ice. Our low temperature experiments were conducted on canonical spin ice materials with a broad range of nuclear spin values. By measuring the magnetic relaxation, or monopole current, we demonstrate strong evidence that dynamical coupling with the hyperfine fields bring the electronic spins associated with magnetic monopoles to resonance, allowing the monopoles to hop and transport magnetic charge. Our result shows how the coupling of electronic spins with nuclear spins may be used to control the monopole current. It broadens the relevance of the assisted quantum tunnelling mechanism from single molecular spins to emergent excitations in a strongly correlated system.

[1] Institut Néel, C.N.R.S—Université Grenoble Alpes, BP 166, 38042 Grenoble, France. [2] School of Physics and Astronomy, Cardiff University, Cardiff CF24 3AA, UK. [3] Clarendon Laboratory, Physics Department, Oxford University, Oxford OX1-3PU, UK. [4] Kyushu Institute of Technology, Kitakyushu 804-8550, Japan. [5] Department of Physics, University of Warwick, Coventry CV4 7AL, UK. [6] London Centre for Nanotechnology and Department of Physics and Astronomy, University College London, 17-19 Gordon Street, London WC1H 0AJ, UK. Correspondence and requests for materials should be addressed to C.P. (email: carley.paulsen@neel.cnrs.fr) or to S.R.G. (email: giblinsr@cardiff.ac.uk)

In the canonical dipolar spin ice materials ($Dy_2Ti_2O_7$, $Ho_2Ti_2O_7$)[1–4], rare earth ions with total angular momentum $J = 15/2$ ($Dy^{3+}$) and $J = 8$ ($Ho^{3+}$) are densely packed on a cubic pyrochlore lattice of corner-linked tetrahedra. The ions experience a very strong $\langle 111 \rangle$ crystal field, resulting in two effective spin states ($M_J = \pm J$) that define a local Ising-like anisotropy. At the millikelvin temperatures discussed here (0.08 K < T < 0.2 K), a lattice array of such large and closely spaced spins would normally be ordered by the dipole–dipole interaction[5], but the pyrochlore geometry of spin ice frustrates the dipole interaction and suppresses long-range order. Instead, the system is controlled by an ice-rule, that maps to the Pauling model of water ice[1–4]. In the effective ground state, the spins describe a flux with closed-loop topology and critical correlations, that may be described by a local gauge symmetry rather than by a traditional broken symmetry[6]. This strongly correlated spin ice state is stabilised by a remarkable self-screening of the dipole interaction[7,8]. Excitations out of the spin ice state fractionalise to form effective magnetic monopoles[6,9], but the excited states are no longer self-screened and this manifests as an effective Coulomb interaction between monopoles. The static properties of spin ice are accurately described by the monopole model[10]. The dynamic properties can also be described by assuming an effective monopole mobility[11–13], but there have been few studies of the microscopic origin of the monopole motion[14].

The field and energy scales involved in monopole motion are illustrated in Fig. 1a–e. When a monopole hops to a neighbouring site a spin is flipped (Fig. 1a). For an isolated monopole (far from any others) this spin flip takes place at nominally zero energy cost (Fig. 1b) because contributions from near-neighbour antiferromagnetic superexchange and ferromagnetic dipole–dipole coupling individually cancel. The cancellation of the field contribution relies on the dipolar self-screening[8] that maps the long-range interacting system[7] to the degenerate Pauling manifold of the near neighbour spin ice model[2]. This surprising cancellation is a key result of the many-body physics of spin ice. In practice, a monopole hop may also involve a finite energy change arising from longitudinal fields at the spin site: the main source of fields is nearby monopoles[6] (Fig. 1b), while further contributions arise from corrections to the mapping, which give a finite energy spread to the Pauling manifold[15] (here of order ~0.1 K[16]). The mechanism of the hop is believed to be quantum tunnelling and several key signatures of this have been observed in the high temperature regime between 2 and 10 K[12–14,17,18].

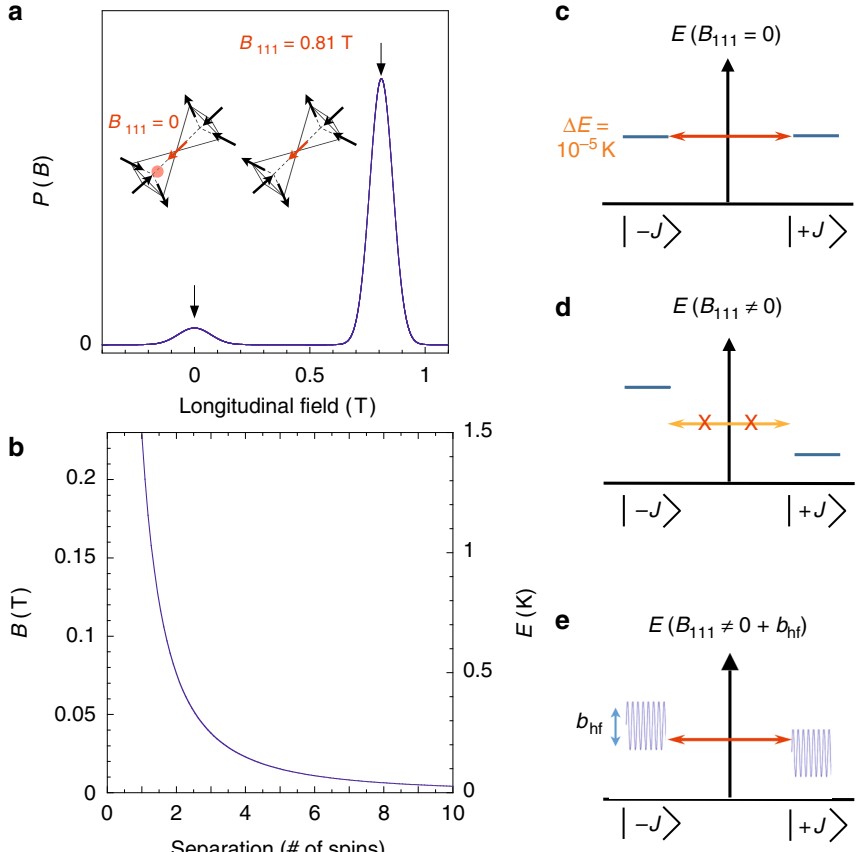

**Fig. 1** How magnetic monopoles tunnel in spin ice. A magnetic monopole is a many-body state that moves via the dynamics of local flippable spins. **a** A qualitative schematic of the longitudinal field distribution ($P(B)$) around a central flippable spin (red), showing how the distribution is centred around zero field (B = 0 T) when there is local monopole (red sphere), and centred around 0.81 T when there is no monopole, which is the case for the vast majority of spins at the millikelvin temperatures discussed here. (Note that (i) the 0 T peak is greatly exaggerated to show on the same scale as the 0.81 T peak; (ii) 0.81 T represents the true field for $Dy_2Ti_2O_7$—including antiferromagnetic exchange reduces the molecular field to about 0.43 T). The broadening of the distribution arises in part from the presence of monopoles and in part from the finite energy spread of Pauling states. **b** The longitudinal field and energy cost of a monopole hop for an isolated monopole–antimonopole pair as a function of the distance between them. **c** The resonant tunnelling process of a flippable spin associated with a monopole; a longitudinal field less than the tunnel splitting for an isolated spin, $\Delta E \approx 10^{-5}$ K[14], will allow tunnelling transitions between the plus and minus spin states. **d** A schematic showing how monopole fields can take the flippable spins off-resonance, such that tunnelling is suppressed. **e** If the spins are not too far off the resonance condition, then rapidly varying hyperfine fields $b_{hf}$ from the precession of nuclear moments can bring otherwise blocked spins into resonance and thus relaxation continues by tunnelling

At lower temperatures ($T < 0.6$ K), spin ice starts to freeze[18]. This is due in part to the rarefaction of the monopole gas whose density $n(T)$ varies as $\sim e^{-|\mu|/T}$ where the chemical potential $|\mu| = 4.35$ and $5.7$ K for $Dy_2Ti_2O_7$ and $Ho_2Ti_2O_7$, respectively[6], and also in part to geometrical constraints that create non-contractable, monopole–antimonopole pairs that cannot easily annihilate[19]. These factors, which are independent of the monopole hopping mechanism, suggest that the relaxation rate $v(T) \propto n(T)$ will fall to exponentially small values at low temperature ($T < 0.35$ K).

Previous thermal quenching experiments have demonstrated monopole populations well below the nominal freezing temperature that are both long lived and able to mediate magnetic relaxation[20]. This paradoxical frozen but dynamical character of the system suggests the relevance of resonant magnetic tunnelling, where magnetisation reversal can only occur when the longitudinal field is smaller than the tunnelling matrix element $\Delta E$. The monopolar fields may add a longitudinal component that takes the spin off the resonance condition (Fig. 1c, d) but in addition may add a transverse component that amplifies $\Delta E$: together these lead to a suppression and dispersion of the monopole mobility.

In the following, we will demonstrate experimentally that hyperfine interactions (Fig. 1e) play a significant role in bringing monopoles back to their resonance condition, enabling dynamics at very low temperatures ($T < 0.35$ K).

## Results

**Samples**. To investigate the effect of nuclear spins on the magnetic relaxation in spin ice, we studied four spin ice samples: $Ho_2Ti_2O_7$, with $I = 7/2$ and three $Dy_2Ti_2O_7$ samples spanning a range of nuclear spin composition from $I = 0$ to $I = 5/2$. Details of nuclear spins and hyperfine parameters are given in Table 1. $Ho^{3+}$ is a non-Kramers ion with intrinsically fast dynamics owing to the possibility of transverse terms in the single-ion spin Hamiltonian, while $Dy^{3+}$, being a Kramers ion, has intrinsically much slower dynamics. However, it should be noted that, at low temperature, bulk relaxation is slower in $Ho_2Ti_2O_7$ than in $Dy_2Ti_2O_7$, owing to its larger $|\mu|$ and hence much smaller monopole density (see Supplementary Fig. 1).

**Thermal protocol**. In previous experiments we have accurately manipulated the monopole density in $Dy_2Ti_2O_7$ by rapid magnetothermal cooling (Avalanche Quench Protocol, AQP) the sample through the freezing transition, allowing the controlled creation of a non-equilibrium population of monopoles in the frozen regime[20]. However, it is more problematic to cool samples containing Ho, due to the large Ho nuclear spin which results in a Schottky heat capacity of 7 J mol$^{-1}$ K$^{-1}$ at 300 mK. Indeed this anomaly has been exploited by the Planck telescope where the bolometers are attached to the cold plate by yttrium–holmium feet thus allowing passive filtering with a several hour time

constant that was crucial to the operation of the system[21]. For $Ho_2Ti_2O_7$ this means difficulty in cooling. Therefore, during some of the runs the sample temperature was recorded via a thermometer directly mounted on the sample face. Figure 2a shows the monitoring of the sample temperature as it approaches equilibrium for $Ho_2Ti_2O_7$ and $Dy_2Ti_2O_7$ during and after the AQP. The inset of Fig. 2a shows that only a few seconds are required to cool the samples from 0.9 to 0.2 K, which is well below the freezing transition. Whereas $Dy_2Ti_2O_7$ continues to cool, reaching 80 mK after only 10 s, $Ho_2Ti_2O_7$ takes nearly 2000 s to reach the same temperature. Hence, the data shown here were taken at 80 mK for $Dy_2Ti_2O_7$ and 200 mK (and 80 mK when possible) for $Ho_2Ti_2O_7$.

**Monopole density**. We have phenomenologically estimated how the monopole density depends upon the rate of sample cooling, $dT/dt$ and the spin relaxation time $\tau(T) = 1/v(T)$, which is derived from the peaks in the imaginary component of the ac susceptibility. Differentiation of $\tau(T)$ to give $d\tau/dT$ and hence $dT/d\tau$, allows definition of an equilibrium cooling rate $dT/d\tau$, that gives the maximum cooling rate that may still maintain equilibrium. Figure 2b compares $dT/dt$ and $dT/d\tau$ for both $Dy_2Ti_2O_7$ and $Ho_2Ti_2O_7$. It can be seen that after the AQP, $dT/dt$ for $Ho_2Ti_2O_7$ crosses the equilibrium curve and goes out of equilibrium at $\approx 0.9$ K, and for $Dy_2Ti_2O_7$ at $\approx 0.72$ K. The upper limit of the monopole density at low temperature can be estimated by equating it to the theoretical value at the crossing temperature: thus we find one monopole on approximately every $10^3$ tetrahedra for both $Ho_2Ti_2O_7$ and $Dy_2Ti_2O_7$.

**Spontaneous relaxation**. We studied the effect of wait time $t_w$ between the end of the avalanche quench and the application of the field with the aim to determine the effect of nuclear spins on the monopole dynamics. Varying the wait time deep in the frozen regime allowed us to gauge the spontaneous evolution of the zero-field monopole density as a function of time: that is, if monopoles recombine in a time $t_w$, then the observed monopole current will be smaller, the longer the wait time. Two separate experiments were designed to study these effects. In the first experiment (Fig. 3) after waiting we applied a constant field and measured the magnetisation $M$ as a function of time. In the second experiment (Fig. 4) we investigated the effect of wait time on the magnetothermal avalanches[22–24] that occur on ramping the field to high values. Both of these allowed access to the magnetic current density $J_m = dM/dt$. Full details of the experimental conditions are given in Supplementary Note 1 and Supplementary Fig. 2.

The monopole current is controlled by multiple factors. In the simplest model[9] there are three of these: the monopole density $n$, the monopole mobility $u$ (related to the spin tunnelling rate) and the bulk susceptibility $\chi$. Thus $J_m = dM/dt = v(M_{eq} - M)$ where $M_{eq} = \chi H$ is the equilibrium magnetisation and $v \propto un$. In general it is difficult to deconvolve these various factors. In ref. [13] it was

**Table 1 Hyperfine properties of the used materials $R_2Ti_2O_7$ ($R =$ Ho, Dy)**

| $R$ | $I$ | $\mu_N$ | $A$ (K) | $B_N$ (T) | $|\mu|$ (K) | $\Delta E$ (K) |
|---|---|---|---|---|---|---|
| Ho | 7/2 | 4.17 | 0.3 | 0.034 | 5.7 | $1 \times 10^{-5}$ |
| $^{161}$Dy ($\approx$19% in $^{nat}$Dy$_2$Ti$_2$O$_7$) | 5/2 | $-0.48$ | 0.0216 | $-0.0039$ | 4.35 | $6 \times 10^{-6}$ |
| $^{163}$Dy ($\approx$25% in $^{nat}$Dy$_2$Ti$_2$O$_7$) | 5/2 | 0.67 | 0.0216 | 0.0054 | 4.35 | $6 \times 10^{-6}$ |
| $^{other}$Dy ($\approx$66% in $^{nat}$Dy$_2$Ti$_2$O$_7$) | 0 | 0 | 0.0216 | 0 | 4.35 | $6 \times 10^{-6}$ |
| $^{162}$Dy | 0 | 0 | 0 | 0 | 4.35 | $6 \times 10^{-6}$ |
| $^{163}$Dy | 5/2 | 0.67 | 0.0599 | 0.0054 | 4.35 | $6 \times 10^{-6}$ |

Here, $I$ is the nuclear spin, $\mu_N$ the nuclear moment, $A$ the hyperfine coupling constant, $B_N$ the effective field due to the nuclear moment at $r = 0.5$ Å, $\mu$ is the monopole chemical potential (conventionally negative, hence we quote $|\mu|$) and $\Delta E$ is the tunnel splitting for a typical transverse field of 0.5 T[14]. The $A = 0.0216$ K coefficient for Dy is the average for $^{nat}$Dy$_2$Ti$_2$O$_7$. Note that about 13% of Ti ions have a nuclear moment giving approximately 20 μT field at the site of a Ho or Dy ion

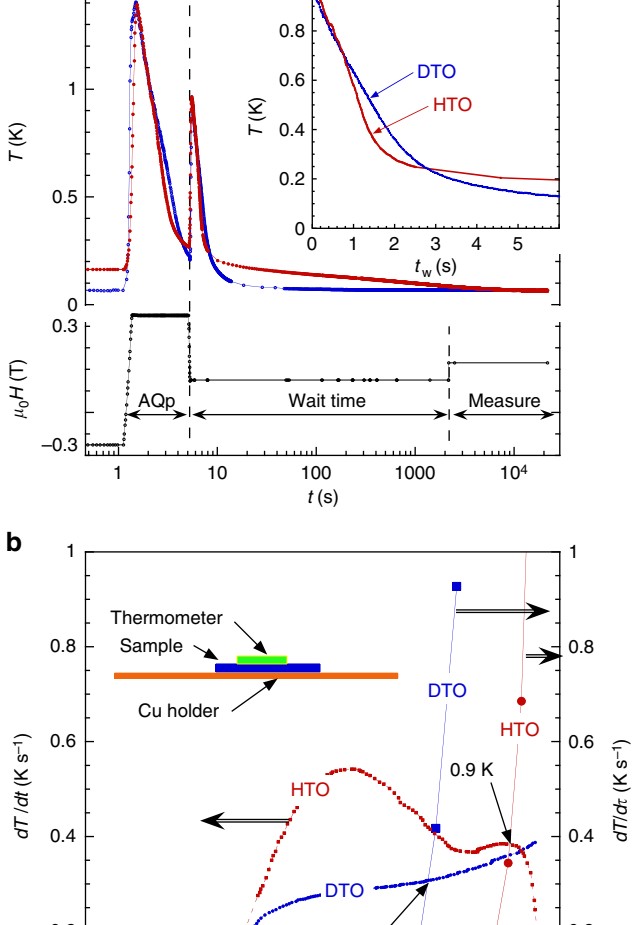

**Fig. 2** Controlled cooling of spin ice below its freezing temperature. How the temperature of the samples varies during and after the AQP: **a** The applied field (black) during an AQP, and the temperatures measured by a small thermometer glued directly on top of the samples (schematically shown in (**b**)) vs. log time for $Ho_2Ti_2O_7$ (HTO, red) and $^{nat}Dy_2Ti_2O_7$ (DTO, blue). The inset shows a zoom of the first 6 s vs. time. **b** Comparison of the sample cooling rates $dT/dt$ as a function of temperature after the AQP for $Ho_2Ti_2O_7$ and $^{nat}Dy_2Ti_2O_7$ from the data in (**a**) to the equilibrium cooling rate $dT/d\tau$ extracted from ac susceptibility data for the two samples (see Supplementary Fig. 6). The cooling rate for $Ho_2Ti_2O_7$ crosses the equilibrium rate at ~0.9 K, and $^{nat}Dy_2Ti_2O_7$ at 0.72 K

achieved by independent measurement of $n(T)$ and $\chi(T)$ to reveal $u(T)$. In the present time-dependent experiments we cannot perform such a direct separation, but by studying $Dy_2Ti_2O_7$ samples with different isotopes, it seems reasonable to assume that the susceptibility and starting density are roughly the same, so the variation in mobility (hop rate) will dominate differences between the samples. Inclusion of $Ho_2Ti_2O_7$ in the comparison gives a further point of reference: the starting monopole densities (see above) and susceptibilities for $Ho_2Ti_2O_7$ are expected to be comparable to those of $Dy_2Ti_2O_7$, while the tunnel splitting (which controls the intrinsic mobility) is also estimated to be of the same order[14] in the appropriate range of internal fields (see Fig. 1 and ref. [14], Fig. 5).

Figure 3 summarises results for the relaxation of the magnetisation $M(t)$ for the different samples, as well as the value of $M(t = 400\,s)$ and the monopole current $J_m(t = 0)$ as a function of wait time, for a constant applied field of 0.08 T. The $Dy_2Ti_2O_7$ samples show a clear progression in wait time effect that correlates strongly with their relative densities of nuclear spin states. Thus the monopoles recombine during the wait period much more effectively the larger the nuclear spin: that is, the larger the nuclear spin the higher the monopole mobility, the faster the recombination, and the fewer the monopoles at the start of the measurement. In Fig. 3e, f, higher mobility means the relaxation curves ($M(t = 400\,s)$ and $J_m(t = 0)$) shift both up and to the left, so a crossover in curves is expected—and this is indeed observed at the longer times. Near to equilibrium a second crossover would be expected (i.e., the equilibrium current density is higher for the highest mobility), but this crossover is clearly very far outside our time window. Hence our $Dy_2Ti_2O_7$ samples are always far from equilibrium.

The effects observed for $Dy_2Ti_2O_7$ are yet more dramatic in $Ho_2Ti_2O_7$, consistent with the $Ho^{3+}$ non-Kramers character, large nuclear spin, and large hyperfine coupling. Relaxation at 200 mK covers more than two orders of magnitude but is practically extinguished for long wait times, showing that excess monopoles spontaneously recombine to eliminate themselves from the sample. The plots indicate that the half life for monopole recombination in $Ho_2Ti_2O_7$ would be approximately 150 s (much shorter than the equilibrium relaxation time) and suggests that equilibrium in the monopole density is reached at long times. Using the above estimate for the initial monopole density $n(t = 0) \sim 10^{-3}$, we recover a nominal equilibrium density of $n_{eq} = 10^{-5}$ (per rare earth atom). Although this estimate is an upper limit it is nevertheless far from the expected equilibrium density, $n_{eq}(T = 200\,mK) \sim 10^{-13}$ (calculated by the method of ref. [10], see Supplementary Fig. 1). It continues to evolve with temperature, being lower by a further order of magnitude at $T = 80$ mK (Fig. 3f). Most likely, the actual equilibrium monopole density is amplified by defects and disorder in the sample.

**Magnetothermal avalanches.** Figure 4 illustrates the effect of $t_w$ on the magnetothermal avalanches. These occur when the injected power ($\mu_0 H \times J_m$) overwhelms the extraction of thermal energy from the sample to the heat bath[23] such that monopoles are excited in great excess as the temperature steeply rises. The faster and more abundant the monopoles, the lower the avalanche field. To obtain the data in Fig. 4, after the AQP and $t_w$, the applied field was swept at a constant rate, 0.02 T s$^{-1}$ up to 0.4 T. If the avalanche field $H_{ava}(t_w)$, is defined as the field where the magnetisation crosses $1\mu_B$ per rare earth ion, ($0.5\mu_B$ for the $^{163}Dy$ sample) then the difference in avalanche field $\Delta H_{ava} = H_{ava}(t_w) - H_{ava}(t_w = minimum)$ allows us to compare the spread of fields for all samples.

Figure 4a, b shows the experimental results for the isotopically enriched $Dy_2Ti_2O_7$ samples at 80 mK, demonstrating a very clear pattern. In general the spread of $H_{ava}(t_w)$ becomes larger, the larger the nuclear spin, showing again that the nuclear spins strongly enhance the monopole mobility. Thus, the $^{162}Dy$ sample (no nuclear spin) shows negligible evolution of the position of the avalanche field. For $^{nat}Dy_2Ti_2O_7$ (shown in Supplementary Fig. 4) the effect is small, while for the $^{163}Dy$ sample (maximum nuclear spin) the effect of $t_w$ can be clearly seen as a steady progression of $H_{ava}(t_w)$ to higher fields for increasing $t_w$ due to the smaller initial monopole density at the start of the field ramp. Also shown in the figure are the curves that result from slow conventional zero field cooling (CC) from 900 to 80 mK (at 1 mK s$^{-1}$) followed by a 1000 s wait period. For the $^{162}Dy$ and $^{nat}Dy_2Ti_2O_7$ samples the

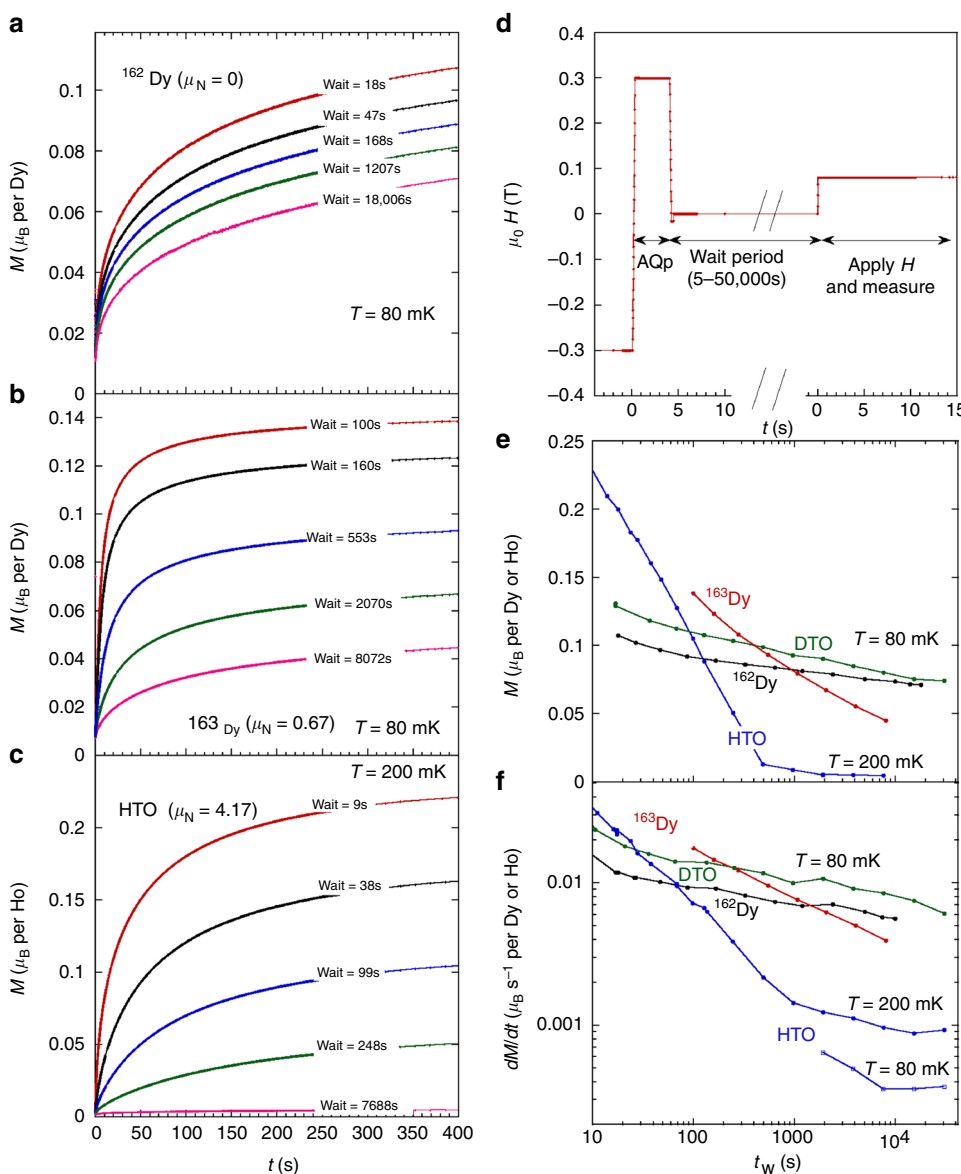

**Fig. 3** Spontaneous evolution of the monopole density during a wait time in zero applied field. This is gauged by the growth of magnetisation ($M$) and monopole current density ($J_m = dM/dt$) after a field is applied; comparison of the different isotopic samples reveals the effect of nuclear spins on the monopole mobility. **a** $^{162}Dy_2Ti_2O_7$ ($^{162}Dy$) and **b** $^{163}Dy_2Ti_2O_7$ ($^{163}Dy$), both measured at $T = 80$ mK, and **c** $Ho_2Ti_2O_7$ (HTO, $T = 200$ mK). $^{nat}Dy_2Ti_2O_7$ (DTO) can be seen in Supplementary Fig. 4. The samples were first prepared using the AQP protocol outlined in (**d**) and discussed further in Methods. After the specified wait periods, a field of 0.08 T was applied and the magnetisation as a function of time was recorded. All measurements shown in the figure were made with the field along the [111] axis; examples for other directions are given in the Supplementary Figs. 10 and 11. **e** Plot of the value of the magnetisation $M$ obtained after the first 400 s for the three samples shown to the left, and for $^{nat}Dy_2Ti_2O_7$ vs. log wait time. Note that the magnetisation values at 400 s remain far from the expected equilibrium value. **f** The monopole current $J_m = dM/dt$ at $t = 0$ for the three samples shown to the left vs. log wait time. Also shown is the monopole current for $^{nat}Dy_2Ti_2O_7$, and the monopole current for $Ho_2Ti_2O_7$ measured at 80 mK

CC avalanche field is offset to higher fields, well outside the distribution of $H_{ava}(t_w)$. For the $^{163}Dy$ sample the CC curves falls within the distribution but near the long wait time curves. Also, we note for $^{163}Dy$, that in a second measurement with better thermal contact, and thus faster cooling during the AQP, the CC curve again falls outside the distribution (shown in Supplementary Fig. 10b). Thus slow cooling is more efficient at approaching equilibrium in $Dy_2Ti_2O_7$ than is the AQP cooling followed by a long $t_w$, especially for the low-nuclear moment samples. This is typical behaviour for frustrated or disordered systems because slow cooling allows the system time to explore all available phase space.

Figure 4c shows a much greater effect of $t_w$ for $Ho_2Ti_2O_7$ with a larger spread of fields, saturating near 0.32 T for the longest $t_w$. This is again consistent with the conclusion that the larger the nuclear spin moment, the more effective the spontaneous monopole recombination. The measurements were performed primarily at 200 mK, but the same conclusion follows from measurements at 80 mK. $Ho_2Ti_2O_7$ also exhibits some unusual behaviour suggesting that the monopole density and magnetisation do not approach equilibrium in a simple way. First, the magnetisation jumps fall short of the $M$ vs. $H$ equilibrium curve taken at 900 mK, even though thermometers placed on the sample indicate that the sample does indeed heat above 900 mK

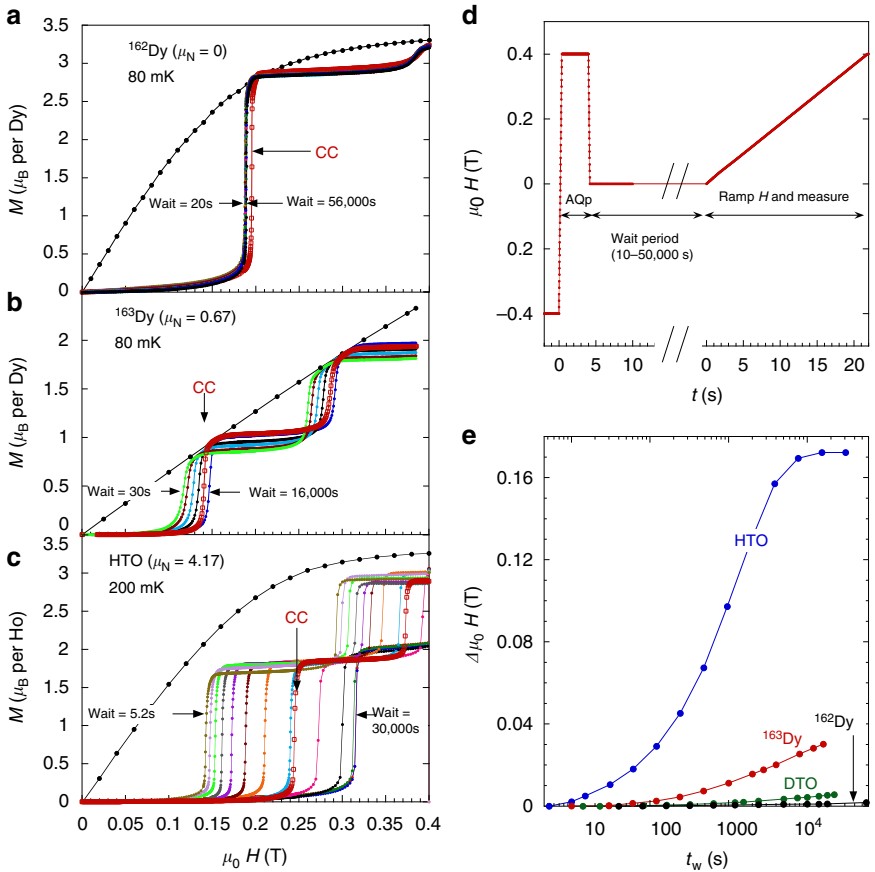

**Fig. 4** Wait time and isotope dependences of magnetothermal avalanches. This gives further evidence of the effect of nuclear spins on the monopole mobility. Avalanches of the magnetisation were recorded while the field was ramped at 20 mT s$^{-1}$ for **a** $^{162}$Dy$_2$Ti$_2$O$_7$ ($^{162}$Dy) and **b** $^{163}$Dy$_2$Ti$_2$O$_7$ ($^{163}$Dy), both measured at $T = 80$ mK, and **c** Ho$_2$Ti$_2$O$_7$ (HTO, measured at $T = 200$ mK). Magnetothermal avalanches for $^{nat}$Dy$_2$Ti$_2$O$_7$ (DTO) can be seen in Supplementary Fig. 4. The samples were first prepared using the AQP and then followed by various wait times (as outlined in (**d**) and discussed in methods) except for the curves marked CC, where the sample was first prepared using the conventional zero field cooled protocol (red squares). Also shown for each of the samples is the equilibrium $M$ vs. $\mu_0 H$ taken at 900 mK (solid black dots). All measurements shown in the figure were made with the field along the [111] axis; examples for other directions are given in the Supplementary Figs. 9–11. **e** Plot of difference in avalanche field $\Delta H_{ava} = H_{ava}(t_w) - H_{ava}$ ($t_w = $ minimum) against log wait time for the data shown in the left as well as $^{nat}$Dy$_2$Ti$_2$O$_7$ (DTO, see Supplementary Fig. 4)

(see Supplementary Note 2 and Supplementary Fig. 3 for more details). Secondly, in contrast to the behaviour of Dy$_2$Ti$_2$O$_7$ discussed above, the CC curve of Ho$_2$Ti$_2$O$_7$ falls in the middle of the distribution of $H_{ava}(t_w)$ indicating, unusually, that waiting long enough at low temperature is an equally efficient way of approaching equilibrium as slow cooling.

## Discussion

The experimental result demonstrated here is that magnetic monopole dynamics in the frozen regime of spin ice are greatly enhanced by the hyperfine coupling of the electronic and nuclear moments. We now argue that this observation finds a natural—albeit surprising—explanation by analogy with the properties of single-molecule magnets[25]. These are metal–organic clusters with large composite spins: some of the most studied include the so called Mn$_{12}$ and Fe$_8$ systems, both of which can be thought of as an ensemble of identical, weakly interacting nanomagnets of net spin $S = 10$ with an Ising-like anisotropy. The degenerate $M_s = \pm S$ states are split by the ligand electric field into a series of doublets. At temperatures smaller than the level separation, the spins flip by resonant tunnelling through a quasi-classical barrier. The signature of a resonant tunnelling effect in Fe$_8$ is a peak in the low temperature relaxation rate around $H = 0$[26]. It quickly became clear that to understand the resonant tunnelling both

dipolar and dynamic nuclear spin contributions to the interactions need to be accounted for. The typical dipolar field in such a system is ≈0.5 K, and the relevant tunnel splitting $\Delta E$ of the order $10^{-8}$ K, meaning that a broad distribution of dipolar field and a static hyperfine contribution would force all the spins off resonance. Prokof'ev and Stamp[27] proposed that dynamic nuclear fluctuations can drive the system to resonance, and the gradual adjustment of the dipole fields in the sample caused by tunnelling, brings other clusters into resonance and allows a continuous relaxation. Hence, the observation of relaxation in single-molecule magnets is fundamentally dependent on the hyperfine coupling with the fields of nuclear spins[28].

The Prokof'ev and Stamp model[27] certainly does not apply in detail to spin ice at low temperatures. First, in single-molecule magnets the spin of any particular complex in the system is available to be brought to resonance, whereas in spin ice, only those spins that are instantaneously associated with a diffusing monopole are available to tunnel (and this presumes that more extended excitations can be neglected). The remaining spins—the vast majority—are, in contrast, static and instantaneously ordered by the ice rules. The rate of flipping of these quasi-ordered spins, which corresponds to monopole pair creation, is negligible at the temperatures studied and the process is not relevant to our experiments. Thus, even at equilibrium, spin ice has an effective

number of flippable spins that depends on temperature (see Supplementary Fig. 1). Away from equilibrium, where our experiments are performed, the number of flippable spins in spin ice further depends on time, with monopole recombination depleting their number. In addition, it seems reasonable to assume that the reduction of the density of monopoles is even more important during the relaxation process; as monopoles move through the matrix magnetising the sample they will annihilate when they encounter a monopole of opposite charge, or become trapped on a defect or on the sample surface. This feature of spin ice is a second important difference with single molecule magnets, as modelled in ref. [27].

A third difference relates to the distribution of internal fields in the system. In spin ice only, the actual field associated with a flippable spin, both before and after a flip, is a monopolar field. Flipping a spin transfers a monopole from site to site (Fig. 1a), dragging the monopolar field with it: a field that is much stronger and of longer range than any conventional dipole field. However, the change in field on a spin flip is dipolar, as in single molecule magnets.

In short, the flippable spins in spin ice are really an aspect of the emergent monopole excitation rather than a perturbed version of an isolated (composite) spin as assumed for the single molecule magnets in ref. [27]. Yet despite this difference, it seems reasonable to suggest that the basic idea of ref. [27] does apply to spin ice. The longitudinal monopolar fields will take flippable spins off resonance (Fig. 1c–e), while the transverse ones will tend to broaden the resonance well beyond the tunnel splitting calculated for an isolated spin, i.e., $\Delta E = 10^{-5}$ K[14]. An applied field can also take flippable spins on or off resonance or broaden the resonance, depending on its direction. Nevertheless, in zero applied field, at very low temperatures we would expect all flippable spins associated with isolated monopoles to be off resonance and hence unable to relax, unless they are brought back to resonance by a combination of the monopole fields and the fluctuating nuclear spins: nuclear assisted flipping of spins will then bring further spins to resonance via the change in dipolar fields, as in the Prokof'ev-Stamp picture[27]. Our experimental results for the wait time dependence of various properties clearly support this proposition: in zero field (during $t_w$) the sample with no nuclear spin is scarcely able to relax its monopole density, while the larger the nuclear spin, the quicker the relaxation. For flippable spins associated with closely spaced monopole–antimonopole pairs the situation is slightly different. Although they are strongly off-resonance (Fig. 1b), the decreasing transition matrix elements will be compensated by the increasing Boltzmann factors required for detailed balance. Also, for the final recombination, a favourable change in exchange energy will reduce the field required to bring spins to resonance (see caption, Fig. 1).

We note in passing that the differences between single molecule magnets and spin ice are also evident in our data. Specifically, a $t^{1/2}$ initial relaxation of the magnetisation is a property of single-molecule magnets, with the $t^{1/2}$ form arising from the dipole interactions[27,29]. Given the very unusual field distribution in spin ice, and the complicating factor of monopole recombination, as described above, it is hardly likely that this functional form will apply. We test for a $t^{1/2}$ decay in the Supplementary Fig. 5 and confirm that it can only be fitted over a narrow time range: to calculate the true time dependence in spin ice poses a theoretical problem.

Our main result has implications for both the theory of spin ice and the theory of nuclear spin assisted quantum tunnelling. First, in previous work[11] we have shown how the low-temperature quenched monopole populations of $Dy_2Ti_2O_7$ obey the nonlinear and non-equilibrium response of monopole theory[30] that was developed assuming a single hop rate. In view of our findings, the theory should apply most accurately to the $Dy_2Ti_2O_7$ sample with no nuclear spins and least accurately to $Ho_2Ti_2O_7$ where the hyperfine splitting energies are of a similar order to the Coulomb energies. In other measurements, presented in Supplementary Figs. 7 and 8, we confirm that this is the case; hence a generalisation of the theory of ref. [30] to include the effect of nuclear spins seems an attainable goal. We also note that $Ho_2Ti_2O_7$ offers the unusual situation that, at low temperatures (<0.35 K) and sufficient wait times, the nuclear spins are ice-rule ordering antiparallel to their electronic counterparts; hence spin ice offers a rare chance to investigate the effect of correlation on nuclear spin assisted quantum tunnelling in a controlled environment. Perhaps this will shed light on some of the unusual properties particular to $Ho_2Ti_2O_7$, as noted above.

Spin ice thus exemplifies a remarkable extension of the concept of nuclear spin assisted quantum tunnelling[27] to the motion of fractionalised topological excitations[6]. This is made possible by the fact that the emergent excitations of the system—the monopoles—are objects localised in direct space that move through flipping spins. As well as illustrating this generic point, our result may also have practical consequences. We have established how coupling with nuclear spins controls the magnetic monopole current and the spectacular magnetothermal avalanches: hence any experimental handle on the nuclear spins of the system would also be a rare experimental handle on the monopole current. Any future application of magnetic monopoles in spin ice will surely rely on the existence of such experimental handles.

## Methods

**Samples**. Single crystals were grown by the floating zone method for all samples, the natural $^{nat}Dy_2Ti_2O_7$ and $Ho_2Ti_2O_7$ samples (DTO, HTO) were prepared at the Institute of Solid State Physics, University of Tokyo, Japan, and $^{162}Dy_2Ti_2O_7$, $^{163}Dy_2Ti_2O_7$ at Warwick University and Oxford University, respectively.

**Measurements**. Measurements were made using a low-temperature SQUID magnetometer developed at the Institut Néel in Grenoble. The magnetometer is equipped with a miniature dilution refrigerator with a base temperature of 65 mK. The fast dynamics after a field change were measured in a relative mode, the slower measurements were made by the extraction method, and the initial relative measurements were adjusted to the absolute value extraction points. The field could be rapidly changed at a rate up to 2.2 T s$^{-1}$.

For all the data shown here the field was applied along the [111] crystallographic direction. Measurements were also performed perpendicular to the [111] direction, as well as along the [001] and [011] directions and on a polycrystalline sample, examples of which are discussed in Supplementary Note 3. In total ten different samples were studied. The direction of the applied field as well as differences in the sample shapes and thermal contact with the sample holder can effect some of the details of the measurements. However, this does not change the main conclusion of the paper: the demonstration of the importance of nuclear assisted quantum tunnelling to the relaxation.

The measurements of temperature vs. time shown in Fig. 2, a bare-chip Cernox 1010-BC resistance thermometer from LakeShore Cryogenics was wrapped in Cu foil and glued on top of the sample as shown in the inset of Fig. 2b.

Cooling $Ho_2Ti_2O_7$ was difficult and warming was also tricky using the AQP, depending on the initial temperatures and wait times. Therefore, to ensure the sample was heated above 900 mK, two AQP were used, separated by 300 s, which explains why the starting temperature for $Ho_2Ti_2O_7$ was higher in Fig. 2 (see Supplementary Note 2 and Supplementary Fig. 3 for further discussions).

A schematic of the AQP used for the preparation of the samples is shown in Fig. 3d. First a field of $-0.3$ T was applied and the sample was allowed to cool to base temperature for 20 min. The field was then reversed at 2.2 T s$^{-1}$ to $+0.3$ T for 4 s then reduced to zero. After a wait period ranging from 10 to 50,000 s, a field of 0.08 T was applied and the relaxation of the magnetisation was recorded. The field $B = 0.08$ T was chosen because it is large enough to get sizeable relaxation, but small compared to the avalanche fields shown in Fig. 4. In this way, when applying the magnetic fields, the relaxation is well behaved and the sample does not heat.

The AQP used for the data of Fig. 4 was similar to the above, except the avalanche field was ±0.4 T. After the wait period the field was ramped at 0.02 T s$^{-1}$, while the magnetisation and temperature of the sample were continuously recorded. For the slow CC protocol measurements shown in Fig. 3, the samples were first heated to 900 mK for 10 s, then cooled at a rate of approximately 0.01 K s$^{-1}$, followed by a waiting period of 1000 s.

## Data availability

Information on the data underpinning the results presented here, including how to access them, can be found in the Cardiff University data catalogue at https://doi.org/10.17035/d.2019.0069144874. The datasets obtained and/or analysed in this study are also available from the corresponding author on reasonable request.

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

## Acknowledgements

S.R.G. thanks Cardiff University for 'seedcorn' funding, and acknowledges the EPSRC for EP/L019760/1 and EP/S016465/1. S.T.B. thanks Patrik Henelius for communicating his independent ideas on nuclear assisted quantum tunnelling in spin ice, and acknowledges the EPSRC for EP/S016554/1. E.L. and C.P. acknowledge financial support from ANR, France, Grant no. ANR-15-CE30-0004. G.B. wishes to thank financial support from EPSRC, UK, through grant EP/M028771/1.

## Author contributions

The experiments were designed and performed by C.P. with inputs and discussions from E.L. and S.R.G. The data were analysed by C.P., E.L., S.R.G. and S.T.B. Contributed materials were fabricated by K.M., D.P. and G.B. The paper was written by C.P., E.L., S.R.G. and S.T.B.

## Additional information

**Competing interests:** The authors declare no competing interests.

