## [Peer Review File · Nature Communications]

Reviewers' comments:

Reviewer #1 (Remarks to the Author):

In this work, C. Paulsen and co-workers investigate the role that nuclear spins are playing in mediating the monopole dynamics in spin ice. In order to systematically address this question, four distinct samples are considered: $A_2\text{Ti}_2\text{O}_7$ with $A = \text{Ho}$ ($I = 7/2$), Dy ($I = \text{mixed}$), and two isotopically enriched samples ^{162}Dy ($I = 0$) and ^{163}Dy ($I = 5/2$). The authors then performed two complementary experiments that probe the monopole dynamics as a function of differing nuclear spin. Both of these experiments utilize the so-called Avalanche Quench Protocol (established by a subset of these authors in a previous work) to cool the samples below their spin freezing temperatures out of equilibrium such that there are large(r) populations of monopole excitations. In the first experiment, the sample is cooled using the AQP and then held at zero field for times between 5 and 50,000 seconds before applying a field and measuring the time dependence of the magnetization. In the second experiment the same AQP and wait protocols are followed, but instead the magnetization is measured as a function of a constantly increasing magnetic field. In both experiments, the samples with the larger nuclear spins are observed to have more rapid re-combination of the monopole pairs.

The question to be addressed in this manuscript is clearly laid out and well-motivated and the experimental approach is well-designed. I find the conclusion to be convincing and they are likely to be of significant interest to those working in the area. Further, I think the comparisons with molecular magnets are interesting and likely to broaden the impact of these results. I also believe that enough experimental details are provided such that the study could be re-produced. I therefore recommend the acceptance of this manuscript for publication in Nature Communications.

I have several comments for the authors consideration.

1. In Figure 1(c) and its caption, a longitudinal field of 10^{-5} K is quoted as the upper-bound for which this tunneling is allowed. The origin of this value is subsequently explained on the last page of discussion but should also be cited in the caption here.
2. The arrows in the inset of Figure 1(a) are, I think, color coded according to whether their longitudinal component is aligned or anti-aligned with the red spin. This is not immediately obvious and could perhaps be explicitly stated in the caption.
3. Page 3, paragraph 3: It is stated that " $\text{Ho}_2\text{Ti}_2\text{O}_7$ is a non-Kramers ion... while $\text{Dy}_2\text{Ti}_2\text{O}_7$ is a Kramers ion". This should say that Ho^{3+} is a non-Kramers, while Dy^{3+} is Kramers.
4. I think the authors should elaborate on why a magnetic field of 0.08 T was chosen for this experiment. Presumably it was selected as it is smaller than the critical field for the first magnetization plateau?
5. On a related note, if you refer to some of the early magnetization measurements on spin ice (for example Matsuhira, Z. Hiroi, T. Tayama, S. Takagi, and T. Sakakibara, J. Phys.: Condens. Matter 14, L559 (2002)), one can see that at $0.08 \text{ T} = 0.8 \text{ kOe}$ the magnetization is approximately $1.5 \mu\text{B}/\text{Dy}$ at 0.4 K for $H \parallel [111]$. Why are the magnetization values in Fig 3 (a-c) an order of magnitude smaller than this?
6. On page 6 it is stated that "...it is far from the expected equilibrium density, $n_{\text{eq}}^{200\text{mK}} \sim 10^{-13}$." Where does this value come from?
7. Can the authors comment on both the ^{163}Dy and the Ho samples (Fig. 4(b) and (c)) show both the $1/6$ and $1/3$ magnetization plateaus while the plateau in ^{162}Dy (Fig. 4(a)) occurs at ~ 2.75

muB?

8. Two typos: "Rarefaction" is misspelled on page 2 and in the caption of Figure 1(e) it should read "far off".

Reviewer #2 (Remarks to the Author):

This paper is an experimental investigation of the dynamics of 4 different "spin ice" systems, viz., $\text{Ho}_2\text{Ti}_2\text{O}_7$, with nuclear spin $I = 7/2$, and three different $\text{Dy}_2\text{Ti}_2\text{O}_7$ samples with nuclear spin values ranging from $I = 0$ to $I = 5/2$. The nuclear spins are key to the investigation, because the authors argue that their results show that the dynamics of the spin ice system relies in a crucial way on the hyperfine coupling to the nuclei. The general idea of nuclear spin-assisted tunneling is of course not new – what is new here is the idea that there is a monopole current/nuclear spin coupling which drives the dynamics of the effective magnetic monopoles in the spin ice system.

Roughly speaking, the evidence for this claim is that the relaxation rate of the monopole density depends strongly on both the magnitude I of the nuclear spins and on the strength of the hyperfine coupling – with higher spin I and/or stronger hyperfine coupling the monopoles recombine faster and the monopole current is higher.

The authors then argue that this behaviour is reminiscent of behaviour seen in molecular magnet systems, where nuclear spins are involved in tunneling relaxation. This paper is not a theory paper, so the arguments given here are rather qualitative, nevertheless, strong parallels are exhibited between the phenomena shown here and those seen in the quantum relaxation of molecular magnet systems.

My feeling is that the experimental results given in this paper are pretty significant – the role of the nuclear spins in the relaxation is convincingly demonstrated - and for this reason alone the paper is worth publishing. I think the paper also gains from the arguments showing the parallel with molecular magnetic relaxation, which suggests a physical mechanism involved. So I recommend publication. However there are a number of things that need to be fixed:

(i) The exposition in the paper is not always terribly clear. I found it quite hard to read the description of the experiments – the text wandered quite a bit, and could be made much more concise, and the discussion was extremely qualitative in parts. I think this discussion would benefit if a clear separation was made between the description of the experiments – which could be more organized, with a more quantitative and precise description of what was found – and the conclusions to be found in the results. The separation between the experimental discussion and the discussion of why one should compare the results with similar phenomena seen in magnetic molecules was however nicely done.

(ii) Given that a parallel is being drawn between the spin ice and magnetic molecule relaxation, which involve 2 different research communities, we need a good description of the relevant physics for each system. The authors have tried to do this, and done a reasonable job, but a few things are a little unclear, notably (a) how do the authors think the form of dipolar interaction between monopoles should be modeled, and where exactly is tunneling entering in the monopole dynamics, and what do they think are the tunneling amplitudes (this is crucial for the comparison with magnetic molecules); and (b) are they proposing a similar interplay between hyperfine and dipolar interactions as that they describe for the magnetic molecules?

(iii) The discussion of the square root law seems to be off the mark. This square root in the

theory for molecular magnet relaxation is not coming from the nuclear dynamics but the long-range dipolar interactions. So if one looks for a square root in the spin ice system relaxation, it needs to be associated with the dipolar interactions between the monopoles. It would presumably break down once the concentration of monopoles that had made transitions reached a certain value, but this is just a speculation until there is some sort of theory, which this paper is clearly going to stimulate. For this reason I suggest that the authors comment on this in a little more detail, with due attention to the timescale when the square root breaks down.

I am also puzzled that the authors did not pay more attention to the field dependence of their results. Given, as shown in Tomesello et al., (their ref 16) that the tunneling amplitudes should depend strongly on applied field, why did they not look at the dependence of monopole relaxation on the same? I appreciate that they have already presented a strong case for nuclear spin mediated tunneling relaxation in this paper, but surely this would strongly reinforce their case. Did they do such measurements? Lack of such measurements does not affect my recommendation, which is that this paper should be published once points (i)-(iii) are dealt with.

Reviewer #3 (Remarks to the Author):

The manuscript 'Nuclear spin assisted quantum tunnelling of magnetic monopoles in spin ice' describes a series of experiments to investigate interactions between spin-ice monopole excitations and the nuclear spin. The results are compared with similar phenomena in molecular magnetic systems. The paper is well written and describes the concepts clearly, however some key pieces of information are not as accessible in the main text as I would have liked. I have detailed the points I would like the authors to address below. Once these are addressed I would consider this manuscript suitable for publication in Nature Communications.

- the direction and magnitude of the applied field used in the measurements should be indicated in the figure captions.

- in the methods it states that the all tested field directions ([111], [001] and [110]) gave the same conclusions. It doesn't state if there is an impact on the nuclear assisted quantum tunnelling and only data for field [111] is shown in the text and SI. Could representative data for the other two field directions be shown in the SI.

- in a number of places in the text the authors refer to 'low temperatures' but don't specify what range they mean. It would be helpful for the reader if an upper bound were to be specified, e.g. in the second sentence of the first paragraph, and at the end of the first complete paragraph on page 3.

- Further justification for how comparisons between the results from Ho₂Ti₂O₇ and Dy₂Ti₂O₇ are made is required. Within the isotopically enriched Dy₂Ti₂O₇ measurements it is easy to justify, however in Ho₂Ti₂O₇ there are significant differences in both the sample (chi, non-Kramers ion, larger nuclear spin) and measurement conditions (temperature, monopole density). How these differences were accounted for prior to drawing comparisons across all samples should be addressed more extensively in the text and any support evidence included in the SI.

We thank the reviewers for their time and effort in reviewing our paper, and we are pleased all three reviewers are positive about our results. We address each question raised by the reviewers in turn and answer the concerns and comments which are reflected in the manuscript as described below.

Reviewer #1 (Remarks to the Author):

In this work, C. Paulsen and co-workers investigate the role that nuclear spins are playing in mediating the monopole dynamics in spin ice. In order to systematically address this question, four distinct samples are considered: $A_2Ti_2O_7$ with $A = Ho$ ($I = 7/2$), Dy ($I = \text{mixed}$), and two isotopically enriched samples ^{162}Dy ($I = 0$) and ^{163}Dy ($I = 5/2$). The authors then performed two complementary experiments that probe the monopole dynamics as a function of differing nuclear spin. Both of these experiments utilize the so-called Avalanche Quench Protocol (established by a subset of these authors in a previous work) to cool the samples below their spin freezing temperatures out of equilibrium such that there are large(r) populations of monopole excitations. In the first experiment, the sample is cooled using the AQP and then held at zero field for times between 5 and 50,000 seconds before applying a field and measuring the time dependence of the magnetization. In the second experiment the same AQP and wait protocols are followed, but instead the magnetization is measured as a function of a constantly increasing magnetic field. In both experiments, the samples with the larger nuclear spins are observed to have more rapid re-combination of the monopole pairs.

The question to be addressed in this manuscript is clearly laid out and well-motivated and the experimental approach is well-designed. I find the conclusion to be convincing and they are likely to be of significant interest to those working in the area. Further, I think the comparisons with molecular magnets are interesting and likely to broaden the impact of these results. I also believe that enough experimental details are provided such that the study could be re-produced. I therefore recommend the acceptance of this manuscript for publication in Nature Communications.

We thank the Reviewer for the positive assessment and comments.

I have several comments for the authors consideration.

1. In Figure 1(c) and its caption, a longitudinal field of 10^{-5} K is quoted as the upper-bound for which this tunneling is allowed. The origin of this value is subsequently explained on the last page of discussion but should also be cited in the caption here.

We have introduced a line commenting on the tunnel splitting for an isolated spin. The caption now contains the following inclusive sentence:

"... a longitudinal field less than the tunnel splitting for an isolated spin, $\Delta E \approx 10^{-5}$ K(14), will allow tunneling.."

2. The arrows in the inset of Figure 1(a) are, I think, color coded according to whether their longitudinal component is aligned or anti-aligned with the red spin. This is not immediately obvious and could perhaps be explicitly stated in the caption.

We apologise for the lack of consistency in the labelling. We have changed the representation to avoid confusion, and have put all the spins in black, except the red central spin. The cases of a monopole on a neighbouring tetrahedron and one far away correspond to the cases of $B = 0$ T and $B = 0.81$ T respectively. We have redrawn the diagram as appropriate and changed the label.

3. Page 3, paragraph 3: It is stated that “ $\text{Ho}_2\text{Ti}_2\text{O}_7$ is a non-Kramers ion... while $\text{Dy}_2\text{Ti}_2\text{O}_7$ is a Kramers ion”. This should say that Ho^{3+} is a non-Kramers, while Dy^{3+} is Kramers.

We thank the reviewer for spotting this mistake and have corrected it as the reviewer suggests.

4. I think the authors should elaborate on why a magnetic field of 0.08 T was chosen for this experiment. Presumably it was selected as it is smaller than the critical field for the first magnetization plateau?

As the reviewer suspects this is a compromise, obviously the field has to be well away from the field (~ 0.15 T) for the first magnetization plateau, but large enough to observe some magnetization once the monopoles begin to annihilate deep in the frozen regime. In addition, as can be seen when compared to figure 4, no sample heating occurs as demonstrated by the magneto-thermal avalanche, in the sample at this low field.

We have modified the text in the methods section to read “*The field $B = 0.08$ T was chosen because it is large enough to get sizable relaxation, but small compared to the avalanche fields shown in Fig. 4. In this way, when applying the magnetic fields, the relaxation is well behaved and the sample does not heat.*”

5. On a related note, if you refer to some of the early magnetization measurements on spin ice (for example Matsuhira, Z. Hiroi, T. Tayama, S. Takagi, and T. Sakakibara, J. Phys.: Condens. Matter 14, L559 (2002)), one can see that at 0.08 T = 0.8 kOe the magnetization is approximately 1.5 $\mu\text{B}/\text{Dy}$ at 0.4 K for $\text{H}||[111]$. Why are the magnetization values in Fig 3 (a-c) an order of magnitude smaller than this?

The long-time magnetization in Fig. 3 is still very far from the thermal equilibrium value: one would have to wait for much longer times to see the values reported in the work of Matsuhira *et al* described above. However it is unlikely that Matsuhira *et al.* reached equilibrium at 0.4 K either, so the actual values of magnetization in this regime are history dependent. At these low temperatures, what happens at equilibrium is, experimentally, a meaningless question: the thrust of our work is that AQP allows meaningful experimental exploration of the low temperature regime as it creates

controlled and reproducible non-equilibrium conditions.

We have attempted to make this point clearer by changes in the text, Firstly, in the “*Spontaneous relaxation*” section we stress we are in a frozen non-equilibrium regime:

“Varying the wait time deep in the frozen regime allowed us to gauge the spontaneous evolution of the zero-field monopole density...”

Secondly in the caption of Fig. 3 we write

“Note that the magnetization values at 400 s remain far from the expected equilibrium value.”

6. On page 6 it is stated that “...it is far from the expected equilibrium density, $n_{eq}^{200mK} \sim 10^{-13}$.” Where does this value come from?

The value was calculated by the iterative method described in detail Ref. 10, we now show the temperature dependence of this calculated monopole density in the Supplementary information, and we add:

“(calculated by the method of Ref. $\text{\cite{Kaiser}}$, see supplementary Fig. 1))”

The calculation of Kaiser et al. is essentially exact for the monopole model of spin ice: there may of course be corrections to the model, but in Ref. 10 these are shown to be very small.

7. Can the authors comment on both the 163Dy and the Ho samples (Fig. 4(b) and (c)) show both the 1/6 and 1/3 magnetization plateaus while the plateau in 162Dy (Fig. 4(a)) occurs at $\sim 2.75 \mu B$?

The values of the magnetization on the plateaus do not represent an equilibrium magnetization so cannot be described as 1/3, 1/6 etc.: instead, as shown in Fig 4b and 4c, the absolute value of the plateau depends upon the starting monopole density and waiting time. It also depends on the thermal contact with the bath and the rate of extraction of heat generated during the ramping process: hence it is not a property of the spin ice sample alone.

8. Two typos: “Rarefaction” is misspelled on page 2 and in the caption of Figure 1(e) it should read “far off”.

We have corrected these typos and thank the reviewer for careful reading of our paper.

Reviewer #2 (Remarks to the Author):

This paper is an experimental investigation of the dynamics of 4 different “spin ice” systems, viz., $\text{Ho}_2\text{Ti}_2\text{O}_7$, with nuclear spin $I = 7/2$, and three different $\text{Dy}_2\text{Ti}_2\text{O}_7$ samples with nuclear spin values ranging from $I = 0$ to $I = 5/2$. The nuclear spins are key to the investigation, because the authors argue that their results show that the dynamics of the spin ice system relies in a crucial way on the hyperfine coupling to the nuclei. The general idea of nuclear spin-assisted tunneling is of course not new – what is new here is the idea that there is a monopole current/nuclear spin coupling which drives the dynamics of the effective magnetic monopoles in the spin ice system.

Roughly speaking, the evidence for this claim is that the relaxation rate of the monopole density depends strongly on both the magnitude I of the nuclear spins and on the strength of the hyperfine coupling – with higher spin I and/or stronger hyperfine coupling the monopoles recombine faster and the monopole current is higher.

The authors then argue that this behaviour is reminiscent of behaviour seen in molecular magnet systems, where nuclear spins are involved in tunneling relaxation. This paper is not a theory paper, so the arguments given here are rather qualitative, nevertheless, strong parallels are exhibited between the phenomena shown here and those seen in the quantum relaxation of molecular magnet systems.

My feeling is that the experimental results given in this paper are pretty significant – the role of the nuclear spins in the relaxation is convincingly demonstrated - and for this reason alone the paper is worth publishing. I think the paper also gains from the arguments showing the parallel with molecular magnetic relaxation, which suggests a physical mechanism involved. So I recommend publication.

We thank the reviewer for the positive assessment and constructive comments.

However there are a number of things that need to be fixed:

- (i) The exposition in the paper is not always terribly clear. I found it quite hard to read the description of the experiments – the text wandered quite a bit, and could be made much more concise, and the discussion was extremely qualitative in parts. I think this discussion would benefit if a clear separation was made between the description of the experiments – which could be more organized, with a more quantitative and precise description of what was found – and the conclusions to be found in the results. The separation between the experimental discussion and the discussion of why one should compare the results with similar phenomena seen in magnetic molecules was however nicely done

With hindsight, we agree that the paper was somewhat dense. To combat this, we have now introduced sections and sub-sections in an effort to make the work more digestible. The division into sections and subsections bring out the distinction between the experiments, what was found, and the subsequent discussion.

Regarding the qualitative nature of the discussion, we have been quantitative wherever possible: we perhaps didn't make this clear enough in the original, but where we quote numbers and general behavior, we are generally applying quite sophisticated theories that have been tested in great detail. For example, in page 6 we make clear that our equilibrium monopole density estimate come from applying the detailed theory of Ref. 10 by adding a line *“(calculated by the method of Ref. 10)”*, and showing the calculated monopole density as a function of temperature in the SI.

We have modified the beginning of the discussion section to read. *“The novel experimental result demonstrated here is that magnetic monopole dynamics in the frozen regime of spin ice are greatly enhanced by the hyperfine coupling of the electronic and nuclear moments.”*

(ii) Given that a parallel is being drawn between the spin ice and magnetic molecule relaxation, which involve 2 different research communities, we need a good description of the relevant physics for each system. The authors have tried to do this, and done a reasonable job, but a few things are a little unclear, notably (a) how do the authors think the form of dipolar interaction between monopoles should be modeled,

This is an important point that we are pleased of the opportunity to clarify. Spin ice is very unconventional as a dipolar system as the many-body dipole interaction surprisingly creates discrete magnetic monopoles that interact according to the Coulomb law —a stronger and longer-range interaction than the familiar dipole interaction. A first point where this matters is in the discussion in the main body of the text: the local longitudinal field at flippable spin is zero Tesla, a surprising result, but an aspect of the many-body physics of spin ice. This is not the same integrated interaction as in a molecular magnet. We have made this clearer with the following modification to the introduction section, our point being to flag to the unfamiliar reader that they should be aware of some non-intuitive features of spin ice:

“The cancellation of the field contribution relies on the dipolar self-screening~\cite{Isakov} that maps the long-range interacting system~\cite{Gingras} to the degenerate Pauling manifold of the near neighbour spin ice model~\cite{BramwellHarris98}. This surprising cancellation is a key result of the many--body physics of spin ice.”

Later, in the discussion, we address the more direct comparison with single molecule magnets. We have clarified their relation in the following paragraph in the discussion, where we carefully list three points of difference:

“The Prokof'ev and Stamp model~\cite{PS} certainly does not apply {it in detail} to spin ice at low temperatures. First, in single molecule magnets the spin of any particular complex in the system is available to be brought to resonance, whereas in spin ice, only those spins that are instantaneously associated with a diffusing monopole are available to tunnel (and this presumes that more extended excitations can be neglected). The remaining spins -- the vast majority -- are, in contrast, static and instantaneously

ordered by the ice rules. The rate of flipping of these quasi-ordered spins, which corresponds to monopole pair creation, is negligible at the temperatures studied and the process is not relevant to our experiments. Thus, even at equilibrium, spin ice has an effective number of flippable spins that depends on temperature (see Supplementary Fig. 1). Away from equilibrium, where our experiments are performed, the number of flippable spins in spin ice further depends on time, with monopole recombination depleting their number. In addition, it seems reasonable to assume that the reduction of the density of monopoles is even more important during the relaxation process; as monopoles move through the matrix magnetizing the sample they will annihilate when they encounter a monopole of opposite charge, or become trapped on a defect or on the sample surface. This feature of spin ice is a second important difference with single molecule magnets, as modelled in Ref. 27.

A third difference relates to the distribution of internal fields in the system. In spin ice only, the actual field associated with a flippable spin, both before and after a flip, is a monopolar field. Flipping a spin transfers a monopole from site to site (Fig. 1a), dragging the monopolar field with it: a field that is much stronger and of longer range than any conventional dipole field. However, the change in field on a spin flip is dipolar, as in single molecule magnets. “

Finally in the next paragraph we pick up on the last point by adding the line:

“...nuclear assisted flipping of spins will then bring further spins to resonance via the change in dipolar fields, as in the Prokof'ev-Stamp picture~\cite{PS}. ”

and where exactly is tunneling entering in the monopole dynamics, and what do they think are the tunneling amplitudes (this is crucial for the comparison with magnetic molecules);

As mentioned above, monopole dynamics is induced by a spin flipping on a tetrahedron. At low temperature, as discussed in the introduction and shown in Figure 1, the mechanism of spin flip, and so of monopole hopping, is believed to be quantum tunneling.

The tunneling amplitudes are hard to determine in spin ice. Based on previous works about nanoparticles and single molecule magnets, it is known that the tunneling amplitude depends, among others, on the size of the magnetic moment, and on the tunnel splitting. Both Ho and Dy have the same magnetic moment. Regarding the tunnel splitting, we expect it to be smaller for Dy³⁺, which is a Kramers ion, than for Ho³⁺ which is a non-Kramers ion. This is indeed confirmed by the calculations of Tomasello et al. for a single ion. Nevertheless, in spin ice, these ions are exposed to a sizeable transverse field. Tomasello's calculations show that, in the presence of this transverse field, the tunnel splitting is expected to be of the same order of magnitude in Dy and Ho based spin ices. So one should expect the tunneling amplitudes to be in the

same range in both systems.

To determine experimentally the tunnel amplitude, the study of the Landau-Zener effect has been extensively used in molecular magnets. It consists in measuring hysteresis loops, and thus the magnetization reversal rate as function of the field sweeping rate, which allows to obtain the tunneling amplitude. This may be done in spin ices by measuring dilute samples, which would give the single-ion tunneling rate, and then by applying transverse fields to modify the tunneling amplitude, and obtain the actual spin ice tunneling rate. This is however hard experimentally due to the multi axis anisotropy of the spins in spin ices.

Also, we think that ac susceptibility on dilute samples might be a good probe of Tomasello's single ion prediction. Indeed this is work we are pursuing with a novel high frequency susceptometer (arXiv:1810.09559).

We feel that it would be premature to go further in discussing the mechanism of tunneling beyond flagging the interesting work of Tomasello et al. We have added a reference to these calculations in the caption of Fig. 1.

"the tunnel splitting for an isolated spin, $\Delta E \approx 10^{-5} K$ cite{Tomasello},"

and (b) are they proposing a similar interplay between hyperfine and dipolar interactions as that they describe for the magnetic molecules?

We believe that the hyperfine interactions are entirely conventional, so they affect spin properties locally: the point is that this local intervention can remove the barrier for a spin flip by broadening the resonance as discussed in Fig. 1 caption and the Discussion. To clarify the discussion we have added at the start of Fig.1 caption:

"A magnetic monopole is a many-body state that moves via the dynamics of local 'flippable' spins. (a) A qualitative schematic of the longitudinal field distribution ($P(B)$) around a central flippable spin (red), showing how the distribution is centred around zero field ($B = 0 T$) when there is one monopole (red sphere),..."

(iii) The discussion of the square root law seems to be off the mark. This square root in the theory for molecular magnet relaxation is not coming from the nuclear dynamics but the long-range dipolar interactions. So if one looks for a square root in the spin ice system relaxation, it needs to be associated with the dipolar interactions between the monopoles. It would presumably break down once the concentration of monopoles that had made transitions reached a certain value, but this is just a speculation until there is some sort of theory, which this paper is clearly going to stimulate. For this reason I suggest that the authors comment on this in a little more detail, with due attention to the timescale when the square root breaks down.

We appreciate the \sqrt{t} itself form comes from dipole interactions, while the nuclear fields are associated with relaxation parameters. To clarify this we have moved the

paragraph addressing this point earlier in the Discussion.

We prefer not to go into any more detail. We are skeptical whether our fit to \sqrt{t} has any real meaning given the small range of t . But the absence of a broad range of \sqrt{t} behaviour is hardly surprising given the special nature of the dipole interaction in spin ice and other complicating factors. To make all this clear we have written:

“We note in passing that the differences between single molecule magnets and spin ice are also evident in our data. Specifically, a $t^{1/2}$ initial relaxation of the magnetisation is a property of single molecule magnets, with the $t^{1/2}$ form arising from the dipole interactions ~\cite{PS, Pauling-t-half}. Given the very unusual field distribution in spin ice, and the complicating factor of monopole recombination, as described above, it is hardly likely that this functional form will apply. We test for a $t^{1/2}$ decay in the Supplementary Fig. 5 and confirm that it can only be fitted over a narrow time range: to calculate the true time dependence in spin ice poses a theoretical problem.”

I am also puzzled that the authors did not pay more attention to the field dependence of their results. Given, as shown in Tomesello et al., (their ref 16) that the tunneling amplitudes should depend strongly on applied field, why did they not look at the dependence of monopole relaxation on the same? I appreciate that they have already presented a strong case for nuclear spin mediated tunneling relaxation in this paper, but surely this would strongly reinforce their case. Did they do such measurements? Lack of such measurements does not affect my recommendation, which is that this paper should be published once points (i)-(iii) are dealt with.

We agree with the reviewer that the results will be field dependent. Indeed in our previous work (Nature Physics, DOI: 10.1038/NPHYS3704) we performed detailed measurements of the magnetic relaxation as a function of field. This showed an exponential square root dependence on the field indicative of monopoles escaping Coulombic binding. In the present paper we have confirmed that a large nuclear spin distorts the $\text{Exp}[\sqrt{B}]$ dependence of the monopole current. This is shown in Supplementary Fig. 8. However, it is an extra level of complexity to analyse field dependence of tunneling in the presence of the nonlinear and nonequilibrium Coulombic physics that we have already established, and we would rather defer this to the future.

Reviewer #3 (Remarks to the Author):

The manuscript 'Nuclear spin assisted quantum tunnelling of magnetic monopoles in spin ice' describes a series of experiments to investigate interactions between spin-ice monopole excitations and the nuclear spin. The results are compared with similar phenomena in molecular magnetic systems. The paper is well written and describes the concepts clearly, however some key pieces of information are not as accessible in the main text as I would have liked. I have detailed the points I would like the authors to address below. Once these are addressed I would consider this manuscript suitable for

publication in Nature Communications

We thank the reviewer for the positive comments and we address the concerns referred to below.

- the direction and magnitude of the applied field used in the measurements should be indicated in the figure captions.

We have added these precisions in the figure captions.

- in the methods it states that all tested field directions ([111], [001] and [110]) gave the same conclusions. It doesn't state if there is an impact on the nuclear assisted quantum tunnelling and only data for field [111] is shown in the text and SI. Could representative data for the other two field directions be shown in the SI.

When addressing this question, we have realized that the data shown for the ^{163}Dy sample were not for the field applied along the [111] direction, but were in fact measurements made on a second polycrystalline sample of ^{163}Dy . We apologize for this mistake.

We have thus corrected figures 3 and 4 to show the data measured along the [111] direction in the main text. One should note that the ^{163}Dy sample shape was not adapted for measurements along the [111] direction: the sample was badly thermalized due to [111] axis being perpendicular to a flat disk and the associated demagnetizing factor was large, which explains the slow slope of the magnetization curve for this sample.

This has strong consequences on the avalanche quench protocol: due to a poor thermal contact, it is less efficient, and the obtained curves are closer to the conventional cooling conditions. For this reason, we have changed the paragraph discussing the difference between the avalanche fields obtained with the AQP and the CC method:

“ For the ^{162}Dy and ^{163}Dy samples the CC avalanche field is offset to higher fields, well outside the distribution of $H_{\text{ava}}(t_{\text{w}})$. For the ^{163}Dy sample the CC curves falls within the distribution but near the long wait time curves. Also, we note for ^{163}Dy , which happens to have better thermal contact, and thus faster cooling during the AQP, the CC curve again falls outside the distribution (shown in Supplementary Fig. 10). ”

We show in the Supplementary Information data for other directions than [111] as requested by the referee, and have modified the Methods paragraph accordingly.

For all the data shown here the field was applied along the [111] crystallographic direction. Measurements were also performed perpendicular to the [111] direction, as well as along the [001] and [011] directions and on a polycrystalline sample, examples of which are discussed in Supplementary Note 3. In total ten different samples were studied. The direction of the applied field as well as differences in the sample shapes

and thermal contact with the sample holder can effect some of the details of the measurements. However this does not change the main conclusion of the paper: the demonstration of the importance of nuclear assisted quantum tunnelling to the relaxation.

These results show the role of hyperfine coupling in the tunneling process in spin ice, regardless the direction of the applied field. Of course, as stated above, the tunneling amplitudes cannot be obtained quantitatively with these experiments, so we cannot determine the tunneling rates for the different directions of the applied field.

-in a number of places in the text the authors refer to 'low temperatures' but don't specify what range they mean. It would be helpful for the reader if an upper bound were to be specified, e.g. in the second sentence of the first paragraph, and at the end of the first complete paragraph on page 3.

We agree with the reviewers suggestion of clarification of what we mean by “low temperature”. In the text we have now specified $T < 0.35$ K, as suggested by the reviewer.

- Further justification for how comparisons between the results from $\text{Ho}_2\text{Ti}_2\text{O}_7$ and $\text{Dy}_2\text{Ti}_2\text{O}_7$ are made is required. Within the isotopically enriched $\text{Dy}_2\text{Ti}_2\text{O}_7$ measurements it is easy to justify, however in $\text{Ho}_2\text{Ti}_2\text{O}_7$ there are significant differences in both the sample (chi, non-Kramers ion, larger nuclear spin) and measurement conditions (temperature, monopole density). How these differences were accounted for prior to drawing comparisons across all samples should be addressed more extensively in the text and any support evidence included in the SI.

To answer this we have spelled out more carefully the expected differences between HTO and DTO. In the original text we had not been clear that, even though in zero field the non-Kramers/Kramers difference is significant, in the field range of interest, the difference is much less significant, as found by Tomasello et al. in their theoretical work. For this reason a direct comparison of HTO and DTO seems reasonable, despite the differences. The revised passage (p.5) now reads:

“Inclusion of $\text{Ho}_2\text{Ti}_2\text{O}_7$ in the comparison gives a further point of reference: the starting monopole densities (see above) and susceptibilities for $\text{Ho}_2\text{Ti}_2\text{O}_7$ are expected to be comparable to those of $\text{Dy}_2\text{Ti}_2\text{O}_7$, while the tunnel splitting (which controls the intrinsic mobility) is also estimated to be of the same order~\cite{Tomasello} in the appropriate range of internal fields (see Fig.1 and Ref. \cite{Tomasello}, Fig. 5.)”

REVIEWERS' COMMENTS:

Reviewer #1 (Remarks to the Author):

Following comments from myself and two other reviewers, the authors have revised their manuscript. In my opinion, all comments have been satisfactorily addressed and the revised manuscript reads very well. The manuscript in its current form is suitable for publication in Nature Communications.

Reviewer #3 (Remarks to the Author):

The authors have satisfactorily addressed the concerns raised by the reviewers in their revised manuscript. The new format makes the paper much easier to follow, I now consider it suitable for publication in Nature Communications.